# ALM Therapy Promotes Functional and Histologic Regeneration of Traumatized Peripheral Skeletal Muscle

**DOI:** 10.3390/biology12060870

**Published:** 2023-06-16

**Authors:** Nina Sarah Hoeger, Thomas Mittlmeier, Brigitte Vollmar, Ioannis Stratos, Geoffrey P. Dobson, Robert Rotter

**Affiliations:** 1Department of Trauma and Reconstructive Surgery, University of Rostock, 18057 Rostock, Germany; 2Institute for Experimental Surgery, University of Rostock, 18057 Rostock, Germany; 3Department of Orthopaedic Surgery, University of Wuerzburg, 97074 Wuerzburg, Germany; 4Heart and Trauma Research Laboratory, College of Medicine and Dentistry, James Cook University, Townsville, QLD 4811, Australia

**Keywords:** muscle regeneration, apoptosis, ALM solution, muscle crush injury, muscle contraction force, proliferation

## Abstract

**Simple Summary:**

Skeletal muscle injuries are very common in everyday life and especially in sports. In most cases, these injuries can be treated conservatively and rarely require surgical therapy. Adenosine, lidocaine and Mg^2+^ (ALM) is a novel solution that has already achieved promising results in experimental studies. Among other things, ALM is cardioprotective and pulmoprotective and improves oxygenation. Our purpose was to determine whether ALM also has protective effects on traumatized peripheral skeletal muscle. Seventy male Wistar rats were anesthetized and subjected to standardized open skeletal muscle trauma. Standardized contusion was performed with the protection of the neurovascular structures. Subsequently, the experimental animals were randomly assigned to the saline control and ALM group and received intravenous infusions of saline or ALM. After 1, 4, 7, 14 and 42 days, the biomechanical regenerative capacity was examined using incomplete tetanic force and tetany. In addition, muscle tissue was assessed for proliferation and apoptosis characteristics by immunohistochemistry. After ALM therapy, biomechanical force development and cell behavior showed significant benefits. More proliferative cells and fewer apoptotic cells were detectable in ALM-treated animals. This indicates that traumatized skeletal muscle could be positively affected by ALM therapy. ALM could be considered in the future as an adjuvant therapy option in clinical practice.

**Abstract:**

Skeletal muscle trauma is a common injury with a range of severity. Adenosine, lidocaine and Mg^2+^ (ALM) is a protective solution and improves tissue perfusion and coagulopathy. Male Wistar rats were anesthetized and subjected to standardized skeletal muscle trauma of the left soleus muscle with the protection of the neurovascular structures. Seventy animals were randomly assigned to saline control or ALM. Immediately after trauma, a bolus of ALM solution was applied intravenously, followed by a one-hour infusion. After 1, 4, 7, 14 and 42 days, the biomechanical regenerative capacity was examined using incomplete tetanic force and tetany, and immunohistochemistry was used to examine for proliferation and apoptosis characteristics. Biomechanical force development showed a significant increase following ALM therapy for incomplete tetanic force and tetany on days 4 and 7. In addition, the histological evaluation showed a significant increase in proliferative BrdU-positive cells with ALM therapy on days 1 and 14. Ki67 histology also detected significantly more proliferative cells on days 1, 4, 7, 14 and 42 in ALM-treated animals. Furthermore, a simultaneous decrease in the number of apoptotic cells was observed using the TUNEL method. ALM solution showed significant superiority in biomechanical force development and also a significant positive effect on cell proliferation in traumatized skeletal muscle tissue and reduced apoptosis.

## 1. Introduction

Skeletal muscle is constantly exposed to various types of injuries. Skeletal muscle injuries are among the most common injuries in primary care and sports [1,2,3]. In clinical practice, acute therapy includes rest, ice, compression and elevation to minimize hematoma. In combination, nonsteroidal anti-inflammatory drugs are also frequently administered [4,5]. In most cases of minor trauma and limited muscle damage, conservative therapeutic procedures are sufficient and lead to a good functional outcome [6]. Few indications require surgical therapy for skeletal muscle injury [6,7,8,9,10]. Healing of skeletal muscle occurs via a constant repair process that is independent of the etiology of the injury [11,12]. The process is divided into three phases: destruction (1), repair (2), and remodeling (3) [11,12]. First, rupture is followed by necrosis of the myofibers [5]. Hematoma is associated with the immigration of inflammatory cells and phagocytes to degrade the resulting hematoma [11,12,13,14]. In addition, an inflammatory cellular response occurs, activating growth factors, cytokines, and chemokines [5,12]. Second, satellite cells begin repairing the injured myofibers. They differentiate into myoblasts and combine into multinucleated myotubes to fuse with the injured muscle fibers [15]. The connective tissue scar is formed between the muscle fiber stumps, and the injury site is revascularized by ingrowing capillaries [5,16]. In the remodeling phase after myoblast differentiation, the focus is on restoring the functional capacity of the muscle [5]. Due to the production of type I collagen, the tensile strength of the connective tissue scar increases significantly [17,18,19]. The myofibers form more and more branches and, thus, penetrate the scar tissue so that the scar tissue decreases over time [11,20]. Patients suffering from major muscle trauma and delayed or inadequate post-traumatic healing of the injured muscle could benefit from supportive and regenerative therapeutic options.

The individual components of adenosine, lidocaine and Mg^2+^ (ALM) have been well-known and researched for many years. Adenosine is an endogenous nucleoside involved in nucleotide production, adenosine triphosphate turnover, and restoration of imbalances. Lidocaine is a fast Na^+^ channel blocker and is used as a local anesthetic and Class 1B antiarrhythmic. Magnesium is a naturally occurring electrolyte and essential for ionic regulation and cellular bioenergetics. Individually, each plays important roles in metabolism, immunomodulation, inflammation and coagulation [21]. ALM solution demonstrated increased survival in hemorrhagic shock, cardioprotective effects, improved microvascular perfusion with concomitant improved oxygenation, and pulmoprotective and nephroprotective properties in small and large animal studies conducted to date [22,23,24,25]. Furthermore, Morris et al. demonstrated less arthrofibrosis with ALM therapy in an animal model with total knee arthroplasty [26]. ALM therapy has also demonstrated reduced systemic inflammation, thrombocyte dysfunction, and coagulopathy after different trauma states [27,28]. More recently, ALM therapy has been shown to preserve hepatocyte architecture with less inflammation and necrosis 3 days after surgical resection, hemorrhage, and shock [29]. In addition, ALM induced cellular quiescence in the surgical margin, which may be a strategy for improved barrier protection and healing [29]. Importantly, the drug combination is unique; Adenosine, Lidocaine or Mg^2+^ alone do not confer these benefits [21]. Therefore, we hypothesized that the ALM solution might improve peripheral skeletal muscle regeneration and functionality after standardized open crush injury. 

## 2. Materials and Methods

### 2.1. Animal Model and Experimental Groups

Seventy male Wistar rats (250–300 g body weight; 8 weeks old; Charles River Laboratories, Sulzfeld, Germany) were used for the experiments. The animals received water and standard laboratory chow ad libitum. Experiments were performed in accordance with the German Animal Welfare Act (LALLF M-V/TSD/7221.3-1-057/16).

Rats were intraperitoneally anesthetized with ketamine 10% (90 mg/kg bw, bela-pharm, Vechta, Germany) and xylazine 2% (25 mg/kg bw, Rompun^®^, Bayer Healthcare, Leverkusen, Germany), and the left hind leg was first shaved and disinfected with povidone–iodine (Betaisodona^®^, Mundipharma, Frankfurt am Main, Germany). Subsequently, the skin of the left hind leg was incised posterolaterally at the level of the Achilles tendon insertion approximately 3.5 cm longitudinally in the direction of the knee joint. The fasciotomy was performed parallel to the skin incision. Afterward, the soleus muscle could be mobilized by blunt dissection from the gastrocnemius muscle, as previously described [30]. The crush injury was performed with an instrumented clamp a total of seven times for 10 s each with a force of 25 N (DMC plus, Hottinger Baldwin Messtechnik GmbH, Darmstadt, Germany). The left soleus muscle was contused in a standardized manner across its entire width from proximal to distal, leaving out the neurovascular structures (Figure 1). The surgical area was post-traumatically lavaged with a sterile 0.9% saline solution. Subsequently, after the crash, a bolus of ALM (composition of ALM: adenosine 2.5 mg/kg/h, lidocaine 5 mg/kg/h and Mg^2+^ 2.5 mg/kg/h in 10 mL 0.9% NaCl) was administered (0.5 mL/kg bw) through intravenous access in the tail vein (BD GmbH, NeoflonTM 26G, Heidelberg, Germany) followed by a one-hour infusion (1.3 mL/kg bw) using ALM stock solutions described by Letson and colleagues [28]. The control group (Saline) received equivalent amounts of saline. The superficial muscle and the skin were sutured using 4-0 Vicryl sutures (Johnson & Johnson Medical GmbH Ethicon, Norderstedt, Germany) in each case. A heating plate was used to keep the body temperature constant between 36–37 °C throughout the operation (Klaus Effenberger, Medizintechnische Geräte, Pfaffing, Germany). After awakening from anesthesia, the animals remained in individual cages with free access to water and standard laboratory chow. For postoperative pain therapy, metamizole was added to the drinking water (375 mg per 100 mL drinking water, daily change, Novaminsulfon Lichtenstein 500 mg/mL, Zentiva Pharma, Berlin, Germany). The ALM group (*n* = 35) and the control group (*n* = 35) were examined after 1 (*n* = 7), 4 (*n* = 7), 7 (*n* = 7), 14 (*n* = 7), and 42 (*n* = 7) days. For immunohistochemical analysis of muscle cell proliferation, animals received a single intraperitoneal injection of BrdU (50 mg/kg) 48 h before the final experiments on days 1, 4, 7, 14, and 42 [31,32,33]. Based on group classification, the group evaluated on day one was injected with BrdU immediately after crush injury, 24 h before evaluation.

### 2.2. Muscle Strength Measurement

To assess muscle strength *in vivo*, as described by Matziolis et al. [30], the animals were intraperitoneally re-anesthetized with ketamine 10% (90 mg/kg bw, bela-pharm, Vechta, Germany) and xylazine 2% (25 mg/kg bw, Rompun^®^, Bayer Healthcare, Leverkusen, Germany) and placed on the heating plate for the duration of the study (*n* = 7 per group and time point). After bilateral exposure of the sciatic nerve and soleus muscle, the Achilles tendon was cut and interspersed with a reinforcement suture (4-0 Vicryl sutures, Johnson & Johnson Medical GmbH Ethicon, Norderstedt, Germany). Afterward, the fixation of the respective hind leg was performed at the knee joint and at the distal lower leg, and the reintwitched Achilles tendon was clamped in the measuring apparatus (NM-01, Experimetria, Budapest, Hungary). The muscle strength measurement was first performed on the healthy right soleus muscle and directly followed by the strength measurement of the traumatized left soleus muscle. The force measurement took only a few minutes per side. The muscle was only kept moist with isotonic saline during the muscle strength measurement. For indirect stimulation of the soleus muscle, the ipsilateral sciatic nerve was stimulated using a curved stimulation electrode (CRS-ST-02-0, Experimetria, Budapest, Hungary) (Figure 2). For incomplete tetanic force stimulation with 9 mA/75 Hz, bipolar was applied five times for 0.1 s each time at an interval of 5 s; for tetany, stimulation was applied at 9 mA/75 Hz five times for 3 s each time at an interval of 5 s (Figure 2 and Figure 3). The contraction forces of complete and incomplete tetany were analyzed by calculating the mean of the maximum values from the first five contractions and expressed as a percentage of the corresponding values of the contralateral healthy muscle (Figure 3). At the end of the experiments, muscle tissue was resected for subsequent histochemistry and immunohistochemistry (see below). 

### 2.3. Histochemistry and Immunohistochemistry

The muscle tissue was fixed in formalin 4% for 2–3 days, then embedded in paraffin and sectioned longitudinally from proximal to distal in 4 µm width. This procedure ensured the assessment of uninjured muscle tissue, injured muscle tissue and the muscle tissue immediately next to the injury zone (penumbra, see Figure 4). The penumbra, defined as the area immediately adjacent to the crush injury, was evaluated by light microscopy at 400× magnification in 10 fields of view per preparation. To evaluate whether intravenous ALM therapy has a positive effect on the number of proliferating cells, the muscle tissue was examined histologically using the immunohistochemical stains BrdU and Ki67. Proliferating BrdU-positive cell nuclei located at the border between muscle fiber and interstitial space were also quantitatively evaluated. These cell nuclei are declared as transitional proliferating cell nuclei in this study. Ki67 is an immunohistochemical marker that shifts from the interior of the nucleus to the surface of the chromosome during the active phase of the cell cycle during mitosis. Because of this property, the Ki67 protein is a good marker to represent the growth fraction of a cell population [34]. Using the Ki67 technique, it is possible to identify nuclear antigens expressed in all phases of the cell cycle (G_1_-, S, G_2_-, and M-phases), with the exception of G_0_-phase [35,36,37]. Similarly, a strong correlation has been demonstrated between the BrdU and Ki67 techniques, and it has been confirmed that Ki67 can reliably detect growth fraction in paraffin-embedded rat tissues [38,39,40,41,42]. Cell apoptosis was quantified by the TUNEL method.

To demonstrate the presence of BrdU in the muscle tissue, we employed a monoclonal mouse anti-BrdU antibody (1:50 dilution; No. M0744, Dako Cytomation, Hamburg, Germany) for Ki67 staining a monoclonal rabbit anti-Ki67 antibody was used (1:200 dilution; No. 15580, Abcam, Berlin, Germany). For visualization, 3,3′-diaminobenzidine was utilized as the chromogen. The sections were examined using a light microscope (BX 51, Olympus, Hamburg, Germany) with a ×40 objective lens (numerical aperture 0.65). The resulting data were presented as the number of cells per square millimeter (mm^2^). Apoptotic cells were detected using the ApopTag kit (ApopTag, S7101, Merck Millipore, Temecula, CA, USA) by the TUNEL method.

### 2.4. Statistical Analysis

The statistical evaluation with the calculations of the mean values, as well as the standard error (SEM) and the graphical representations of the experimental results, were performed with Sigmaplot 13.0 (Systat Software Inc., San Jose, CA, USA). The respective results are given as mean values ± SEM. If the values were normally distributed (Shapiro–Wilk test), the statistics for the group comparison were determined by two-way analysis of variance (two-way ANOVA). Significance levels are indicated at *p* < 0.05. Because of a lack of normal distribution in the evaluation of apoptotic cells, the Mann–Whitney rank sum test with Bonferroni correction was performed.

## 3. Results

### 3.1. General Observations In Vivo

All animals with open crush injury of the left soleus muscle awoke from anesthesia without complications. Use of the left hind leg was only slightly restricted during the first one to three days. No other signs indicating pain or disease were to be observed. The healthy right hind leg showed no limitations in all force measurements and is used as a reference in incomplete tetanic force and tetany to determine the functional deficit of the traumatized left hind leg.

### 3.2. Muscle Strength Measurement

Measurement of contraction forces at the uninjured soleus muscle of the contralateral hindlimb revealed a mean incomplete tetanic force of 0.55 ± 0.01 N and a mean tetanic force of 0.91 ± 0.02 N (*n* = 70). As a result of the open crush injury to the left soleus muscle, a markedly reduced contractile force was measured on day one, with 18% (ALM, incomplete tetanic force) vs. 19% (Saline, incomplete tetanic force) and 13% (ALM and Saline group, tetany) of the force of the contralateral uninjured, healthy soleus muscle. On day four, the traumatized muscle tissue showed recovery of contractile force to 43% (ALM) and 27% (Saline) for incomplete tetanic force and to 42% (ALM) and 20% (Saline) for tetany compared with the force of the contralateral muscle. During the further observation period, the traumatized muscle tissue showed further improvement in muscle force at day 7 to 62% (ALM), as well as 46% (Saline) for incomplete tetanic force and 47% (ALM), as well as 33% (Saline) for tetanic force compared with the force of the contralateral muscle. By day 14, the force development of the contraction force increased to 70% (ALM, incomplete tetanic force) and 66% (Saline, incomplete tetanic force), respectively, to 56% (ALM, tetany) and 55% (Saline, tetany) in relation to the contraction force of the healthy muscle (Figure 5). 

### 3.3. Muscle Tissue Proliferation and Apoptosis

Quantitative analysis of muscle tissue after open crush showed different kinetics of cell proliferation in the ALM group compared with the control group. In the evaluation of BrdU-positive cells, all proliferative cells reached their maximum value at day 1, with 36.7 ± 2.2 cells per mm^2^ in ALM-treated animals compared to 25.1 ± 2.3 cells per mm^2^ in controls (*p* < 0.05) (Figure 6). On day 4, cell numbers were 28.9 ± 4.0 cells per mm^2^ (ALM) and 23.0 ± 1.4 cells per mm^2^ (Saline). Day 7 showed reduced proliferation behavior with 20.0 ± 3.7 cells per mm^2^ in the ALM group and 18.8 ± 1.3 cells per mm^2^ in controls. At day 14, 24.0 ± 0.9 cells per mm^2^ (ALM) and 13.1 ± 0.7 cells per mm^2^ (Saline) were verifiable. After 42 days, the number of proliferative cells was further regressed in both groups, with 5.9 ± 0.6 cells per mm^2^ (ALM) and 2.5 ± 0.1 cells per mm^2^ in controls (Figure 6).

Transitional cells (e.g., cells located at the border between muscle fibers and interstitial space) were assessed separately. On day 1, significantly more transitional cells were detectable in the ALM group (22.4 ± 1.6 vs. 16.3 ± 0.0). Day 4 also showed a significantly higher number of transitional proliferating cell nuclei in the ALM group compared to the control group (22.3 ± 3.9 vs. 16.5 ± 2.6). In the further course, a reduced proliferation behavior was found on day 7, with 13.4 ± 3.1 in the ALM group and 11.2 ± 1.2 in the control group. After 14 and 42 days, a significant difference between the two groups was again observed (ALM-14: 16.7 ± 0.1 vs. Saline-14: 10.6 ± 0.6 and ALM-42: 4.0 ± 0.3 vs. Saline-42: 2.1 ± 0.1).

The proliferation behavior in the evaluation of Ki67-positive cells was similar to that in the BrdU analysis. However, in contrast to BrdU, there were significantly more Ki67-proliferative cells in the ALM group than in saline controls at all time points (Figure 7). On day 1, 83.4 ± 1.4 cells per mm^2^ (ALM) and 57.8 ± 3.5 cells per mm^2^ (Saline) were detected. Progressive values were seen on day 4 with 87.3 ± 2.5 cells per mm^2^ (ALM) and 63.0 ± 2.3 cells per mm^2^ (Saline). Maximum values were reached on day 7 with 90.8 ± 2.5 cells per mm^2^ (ALM) and 72.1 ± 3.7 cells per mm^2^ (Saline). In the further course, the number of proliferative cells was regressive. After 14 days, 70.0 ± 2.4 cells per mm^2^ (ALM) and 51.1 ± 2.2 cells per mm^2^ (Saline) were measured. The lowest number of proliferative cells was seen after 42 days, with 48.1 ± 2.7 cells per mm^2^ (ALM) and 23.1 ± 0.5 cells per mm^2^ (Saline) (Figure 7). 

Quantification of cell apoptosis in traumatized muscle tissue was performed using the TUNEL method. On days 1, 4, and 7, significantly fewer apoptotic cells were detected in ALM-treated animals than in controls (Figure 8). On day 1, there were 10.2 ± 0.8 cells per mm^2^ (ALM) and 13.0 ± 1.0 cells per mm^2^ (Saline), with numbers increasing to 11.4 ± 0.0 cells per mm^2^ (ALM) and 22.7 ± 0.8 cells per mm^2^ (Saline) by day 4. At day 7, the number of apoptotic cells was regressive, with 7.0 ± 0.3 cells per mm^2^ (ALM) and 12.3 ± 0.3 cells per mm^2^ (Saline) (*p* < 0.05). After 14 days, 4.4 ± 0.1 cells per mm^2^ and 2.7 ± 0.1 cells per mm^2^ were detected in ALM and control groups, respectively. At day 42, there was a decrease in apoptotic cells to baseline physiological levels in both groups, with 0.4 ± 0.1 cells per mm^2^ (ALM) and 0.4 ± 0.0 cells per mm^2^ (Saline) (Figure 8). 

In contrast to the different values of cell numbers at the initial study time points, cell proliferation and cell apoptosis were the same in animals from both groups at the end of the study period. 

Histological analysis was conducted on the intact, healthy soleus muscle to examine its proliferation behavior and apoptosis. The results are summarized in Table 1. Significant differences in proliferation values were measured in BrdU staining. This suggests that altered proliferative activity also occurs on uninjured skeletal muscle with the influence of ALM.

## 4. Discussion

In the present study, we have evaluated ALM solution in an animal model of standardized severe muscle trauma for the first time to the best of our knowledge. Recent studies have identified several effects of ALM that may have a positive impact on polytrauma patients, including improvement in cardiac and hemodynamic function, more rapid correction of coagulopathy, reduction in inflammation, endothelial injury, and infection, and improvement in oxygenation [21,23,27,43,44]. In addition, ALM is thought to have anti-apoptotic effects, which have also been demonstrated in the current study in traumatized skeletal muscle tissue [45]. Therefore, better recovery of muscle function could be shown within the period after trauma induction. 

To better understand the pathophysiology of muscle trauma, appropriate experimental trauma models are needed that can realistically represent the trauma, are standardizable and reproducible and allow for further assessment [46].

### 4.1. Animal Skeletal Muscle–Trauma Model

The trauma model used with the contusion of the soleus muscle is an established and standardized model of open severe soft tissue trauma. The muscle is isolated and traumatized, and the blood and nerve supply remain intact. This is a major advantage of this model, as one can achieve standardized myofiber damage without denervation [46]. The preparation of the muscles of the hind leg and, in particular, the soleus muscle, as well as localization of the sciatic nerve, are surgically easily realizable. Due to these properties, the selected trauma model shows high standardization and reproducibility [30,33]. The study focuses on histomorphological and biomechanical regeneration behavior. In their study, Stratos et al. described the different regeneration behavior of the soleus muscle during a sevenfold contusion of the muscle with an instrumented clamp for 10 s each with a force of 5 N, 25 N or >99 N. The traumatized muscle with a force of 5 N showed complete regeneration after 42 days, while for the greater degrees of trauma, a lower regeneration capacity was demonstrated [33]. For the present study, the contusion with the mean degree of the trauma of 25 N was selected because this tissue traumatization has a comparable regenerative potential and does not show complete convalescence after 42, so potential influences of therapeutic approaches on muscle regeneration can be investigated. Furthermore, in the chosen model, it was possible to keep the trauma intensity and severity constant at all times [33]. Alternatively, a variety of other trauma models are available to analyze the regenerative behavior of skeletal muscle tissue. Muscle regeneration is strongly dependent on the chosen trauma model and the severity of the muscle injury. For example, experimental trauma models are possible in which mechanical, by external force application, a closed muscle trauma is generated. For force application, depending on the model, a hammer, a drop weight, or a pneumatically driven bolt is used [47,48,49,50]. The advantages of these methods are that no wound healing of the skin is necessary, infection is excluded, and the muscle injury can be well transferred to the injury mechanism in humans, because many soft tissue injuries are associated with no injury to the skin [47,48,49,50]. Significant disadvantages of the method are the lack of visual verifiability of the muscle application in order to avoid damage to neighboring structures, such as musculature, vessels and nerves [47]. The trauma model used in this study was chosen because of its standardization and reproducibility in trauma induction, the direct visualization of muscle contusion and the comparable regeneration potential.

### 4.2. Groups and Post-Traumatic Examination Times

In order to evaluate muscle recovery in terms of biomechanical strength and histomorphological parameters, the groups were examined at five different postoperative time points. On a postoperative day 1, the evaluation of group 1 was performed to determine the direct influence of the applied ALM solution on the early histomorphologic regeneration process [33]. Between the third and fifth day after trauma, essential processes of proliferation take place. Therefore, the second time point was defined as the fourth postoperative day [12]. Group 3 was evaluated on day seven and group 4 on day 14 after the trauma induction to evaluate the medium-term course of muscle recovery. Between day seven and day 14, the regenerating muscle cells closed the central zone (the gap between the distracting muscle stumps) of the trauma and the resulting scar tissue becomes denser [12]. The final analysis, 42 days after trauma, is designed to investigate the longer-term effects of the ALM application on the regenerative capacity of the muscle tissue, as already after three weeks, the myofibers have largely fused again, and there is only a few scar tissue between them [12].

### 4.3. Timing, Dosage and Mode of Application of ALM

The timing of the ALM application was chosen immediately after induction of crush injury in order to directly influence the regeneration process. The dosage of ALM solution used in this study was based on previous successful studies in other animal models of the rat [28]. The single bolus administration followed by infusion has been used several times in animal models [28,51,52]. The one-hour duration of the infusion was chosen because the stabilization of vital signs over a period of three to four hours, as in other animal models, was not necessary for this study [51,52]. Higher doses do require more intensive monitoring because of the cardiovascular effects of ALM. In the present study, the intravenous application was chosen, as the effect of ALM has only been investigated in this setting. An intramuscular or intraperitoneal application would also be conceivable for the investigations on the traumatized skeletal muscle, as these modes of application also achieved successful results with some modulators [32,53].

### 4.4. Muscle Strength Measurement

The assessment of muscle regeneration through the contraction force provides information about the functional success of therapeutic approaches [54]. In the present study, it was demonstrated that trauma induction of the left soleus muscle leads to a significant reduction in relative muscle strength. In the post-traumatic course, there was a significant increase in relative muscle strength following ALM application on days 4 and 7, demonstrating the enhanced regenerative capacity of the traumatized skeletal muscle. Several studies demonstrated that muscle strength increased over the course of the study period employing various stimuli of muscle regeneration, as was observed here [32,53].

From previous studies on ALM solution, it is known that small-volume ALM therapy has multiple protective effects on the whole organism, such as the improvement of the cardiac and hemodynamic function, correction of coagulopathy, reduction of inflammation, endothelial injury, and infection, as well as improvement of oxygenation [21,23,27,43,44]. Additionally, the antifibrotic and antiapoptotic effect of ALM therapy may explain the significantly higher strength on days 4 and 7 compared with the control groups [26,45].

### 4.5. Muscle Cell Proliferation and Apoptosis after Crush Injury and ALM

Cell proliferation was detected by two methods. BrdU, which is a standard method and Ki67, which is easier to handle because no *in vivo* administration is required. The number of Ki67-positive cells is much higher than the number of BrdU-positive cells because BrdU detects cells in the S phase of the cell cycle, and Ki67 is expressed throughout the mitotic process (G_1_-, S-, G_2_-, and M-phase) and as a result, approximately 50% more cells are marked than with the BrdU method [55].

Immunohistochemistry is adapted from other studies also performed on peripheral skeletal muscle. Ferreira et al. describe that the cells with labeled nuclei were typed according to their position as satellite cells (peripheral location), central myonuclei, endothelial cells, and interstitial nonendothelial cells (Fibroblasts) [31]. Additionally, Anderson et al. describe satellite cell activation that resulted in DNA synthesis was determined by immunostaining for BrdU-positive nuclei in satellite cells attached to fibers and in cells that had migrated away from fibers [56].

The improved muscle function by ALM was also demonstrated at the histological level. Animals treated with ALM showed significantly higher numbers of transitional proliferating cell nuclei and interstitial cells. Studies performed on the same trauma model with post-traumatic EPO (Erythropoietin) or G-CSF (Granulocyte-Colony Stimulating Factor) application also showed an increased cell proliferation rate in the early postoperative days, especially on days 1 and 4 [32,53]. This is also consistent with the results of this study and shows that the highest proliferation rates are to be expected in the first post-traumatic days. In particular, satellite cells, as myogenic stem cells, are essential for muscle regeneration [57,58]. Until now, the exact mechanism of how ALM exerts proliferative effects in skeletal muscle is not yet known and requires further studies. There are more options for immunohistochemical staining for the detection of satellite cells which had not been applied here, making the direct proof of satellite cells difficult. 

A possible antiapoptotic effect of ALM has been reported previously [22,45]. A clear antiapoptotic effect was also demonstrated in our study, especially on days 4 and 7. This is also consistent with the study conducted with G-CSF, which also showed the maximum of apoptotic cells on day 4 in the control group [53]. Reduced apoptosis after traumatic muscle injury may have important clinical benefits for recovery. However, further studies are needed to explore the exact mode of action.

### 4.6. Limitations of the Present Study

ALM is an innovative combination of active ingredients that have been widely researched in multiple experimental trauma studies. The protective effect of ALM has already been demonstrated in several organ systems [21,59]. Because ALM has been administered intravenously in previous studies, we used this route of administration, which may not be the preferred route for isolated muscle injury. Intraperitoneal or intramuscular application or even multiple applications, as in other studies of open severe soft tissue trauma, might be considered in the future [32,53]. Additionally, complementary and advanced histological methods are needed to further evaluate the effect of ALM on injured peripheral skeletal muscle (e.g., for treatment of skeletal muscle tissue: quickly frozen in liquid nitrogen pre-cooled isopentane or the myogenic markers: Pax7, MyoD and myogenin). Another limitation of the present study is the exclusive use of male rats, with additional studies required to determine sex-specific treatment effects. Additionally, the effects of ALM on other species are of high relevance. Further studies are also needed to explore the mechanisms of the regenerative effect of ALM on traumatized skeletal muscle. Finally, clinical studies are required to prove the efficacy, safety and quality of the ALM solution in order to establish ALM as a therapeutic agent.

## 5. Conclusions

Intravenous ALM therapy in an animal model of the rat hindleg with standardized open severe soft tissue trauma achieved significantly better functional regeneration of skeletal muscle and also induced an increased number of proliferative cells with a concomitant lower number of apoptotic cells compared to saline controls. The observations justify further research in this promising approach to better muscle preservation and function following severe muscle trauma. ALM could be a promising and innovative therapy option that could provide adjuvant support to the post-traumatic regeneration process in everyday trauma surgery.

## Figures and Tables

**Figure 1 biology-12-00870-f001:**
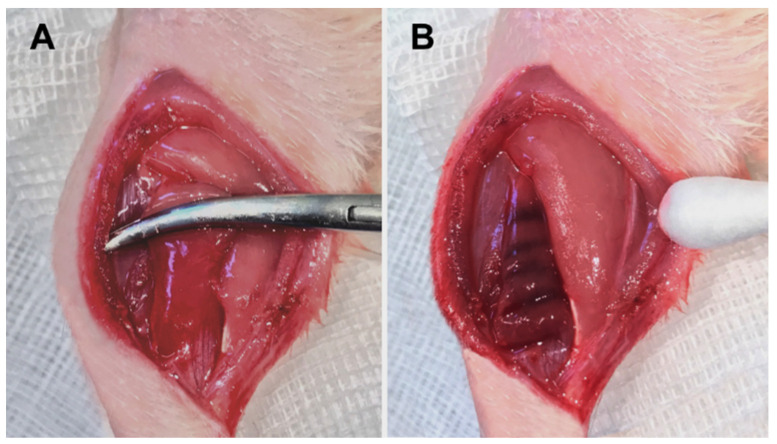
Trauma induction of the left soleus muscle (**A**) and the macroscopic result immediately after the contusion (**B**).

**Figure 2 biology-12-00870-f002:**
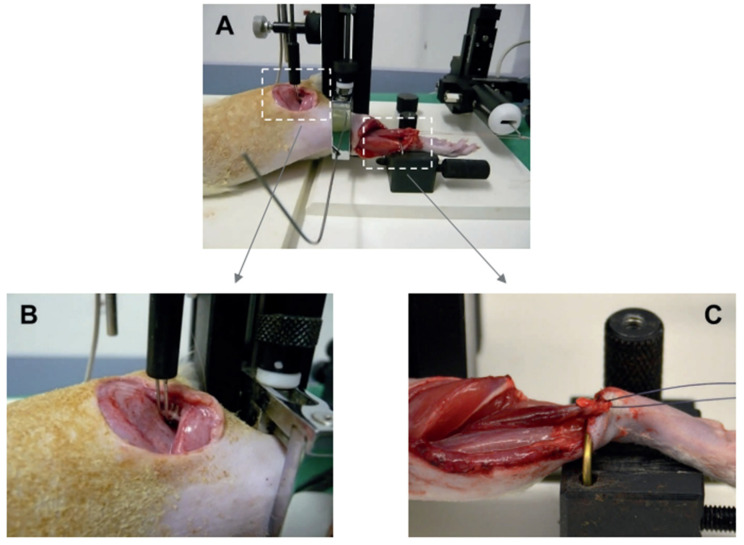
Experimental setting of the biomechanical force measurement (**A**). For indirect stimulation, the sciatic nerve was applied to the stimulation electrode (**B**). The soleus muscle was clamped in the experimental device (**C**). The muscle force was measured by a measuring device and visualized.

**Figure 3 biology-12-00870-f003:**
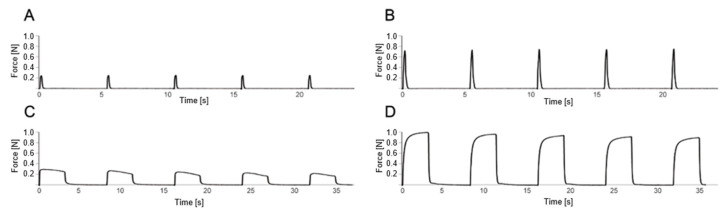
Illustrative data sequences of the muscle force of the soleus muscle of a rat. The x-axis shows the time, and the y-axis the tensile force of the muscle during direct stimulation of the sciatic nerve with 9 mA/75 Hz. The upper field shows 5 incomplete tetanic contractions, each of the injured left soleus muscle (**A**) and the non-injured right soleus muscle (**B**) after stimulation of the sciatic nerve for 0.1 s. The lower field shows the tetany of the injured left soleus muscle (**C**) and the non-injured right soleus muscle (**D**) after 5-fold stimulation of the sciatic nerve for 3 s each. To determine the relative increase in force during incomplete tetanic contraction, the maximum values of 5 continuous incomplete tetanic contractions of the left soleus muscle were averaged (**A**) and divided by the average of the maximum values of 5 continuous incomplete tetanic contractions of the right soleus muscle (**B**). In the same way, the relative increase in force during tetany was calculated (**C**,**D**), and the force of the left soleus muscle was expressed as a percentage of the force of the contralateral side.

**Figure 4 biology-12-00870-f004:**
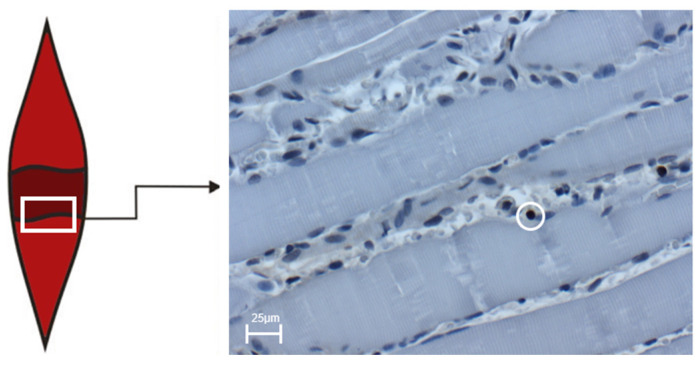
The box shows the transition zone (penumbra zone) between healthy and traumatized tissue of the soleus muscle. The right image shows an exemplary light microscopic image at 400× magnification of BrdU-positive cells in the control group on day four after trauma induction. The cell marked in the circle represents BrdU-positive cells. The scale corresponds to 25 µm.

**Figure 5 biology-12-00870-f005:**
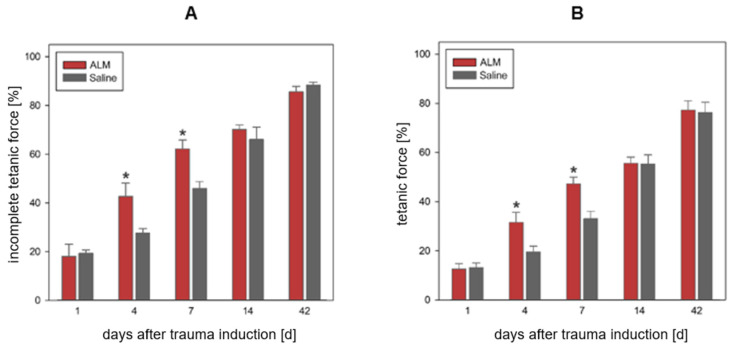
Relative biomechanical force development during incomplete tetanic force (**A**) and tetany (**B**) of the traumatized soleus muscle after indirect stimulation of the sciatic nerve compared to the healthy contralateral muscle. The ALM groups with adenosine, lidocaine and Mg^2+^ (ALM) are shown in red (*n* = 7), and the control groups in gray (*n* = 7). Data are given as mean values ± SEM, * *p* < 0.05. For both incomplete tetanic force and tetany, relative biomechanical force development on days 4 and 7 was significantly higher in the ALM groups than in the control groups.

**Figure 6 biology-12-00870-f006:**
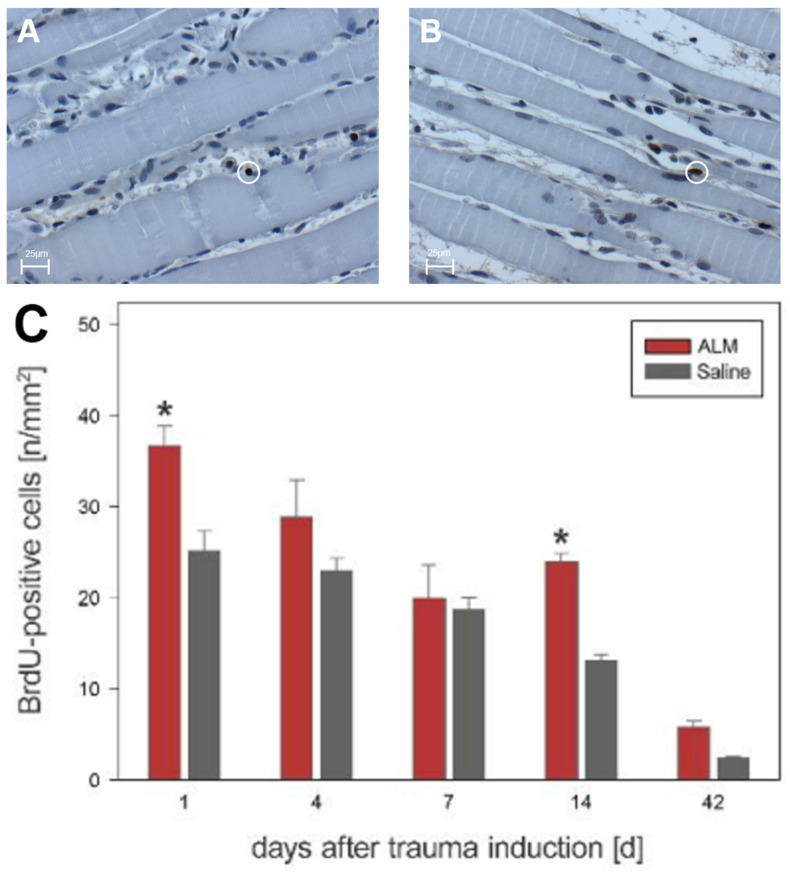
Histological evidence of all BrdU-positive cells in the ALM groups (red, *n* = 7) and control groups (gray, *n* = 7) (**C**). Data are given as mean values ± SEM, * *p* < 0.05. (**A**,**B**) show exemplary light microscopic images at 400× magnification of BrdU-positive cells in the control group (**A**) and ALM group (**B**) on day four after trauma induction. The cells marked in the circle represent BrdU-positive cells. The scale in (**A**,**B**) corresponds to 25 µm.

**Figure 7 biology-12-00870-f007:**
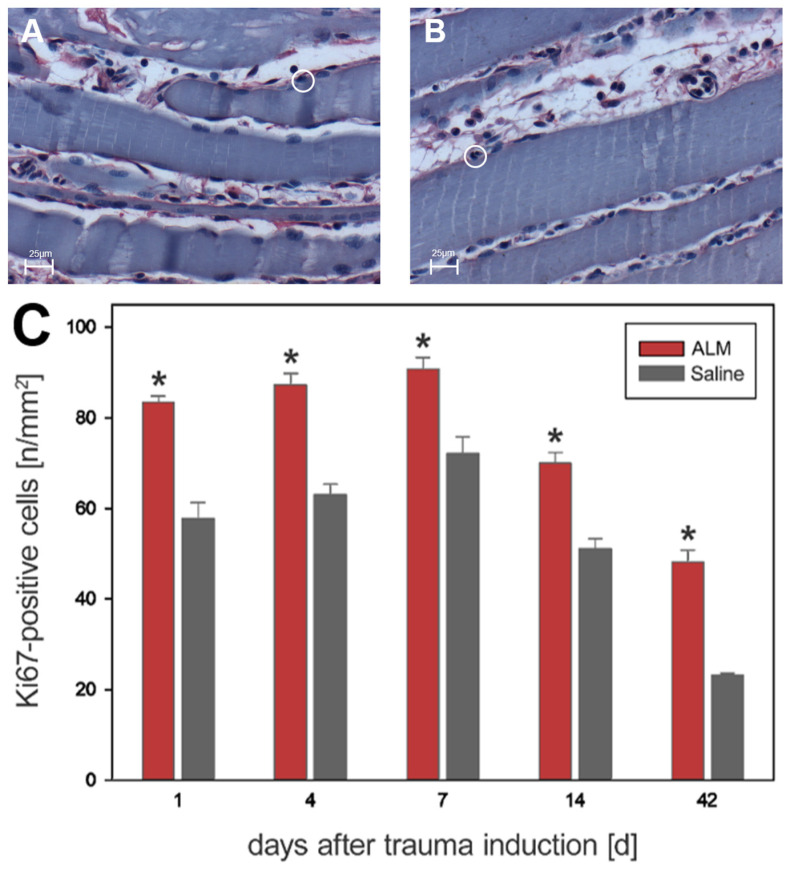
Histological evidence of all Ki67-positive cells in the ALM groups (red, *n* = 7) and control groups (gray, *n* = 7) (**C**). Data are given as mean values ± SEM, * *p* < 0.05. (**A**,**B**) show exemplary light microscopic images at 400× magnification of Ki67-positive cells in the control group (**A**) and ALM group (**B**) on day four after trauma induction. The cells marked in the circle represent Ki67-positive cells. The scale in (**A**,**B**) corresponds to 25 µm.

**Figure 8 biology-12-00870-f008:**
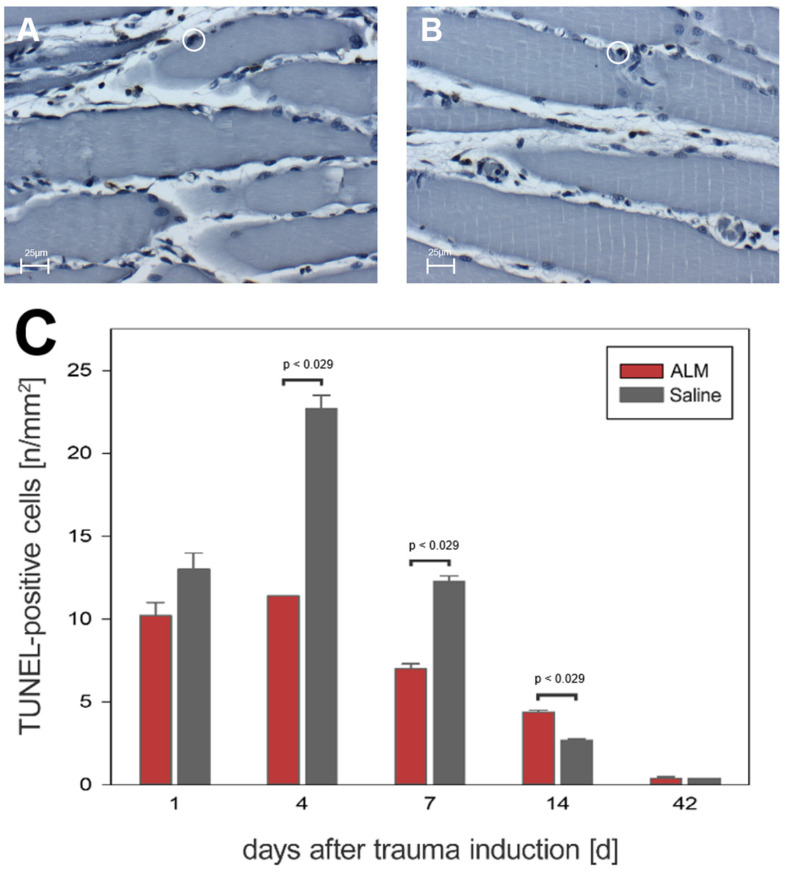
Histological evidence of all TUNEL-positive cells in the ALM groups (red, *n* = 7) and control groups (gray, *n* = 7) (**C**). Data are given as mean values ± SEM, Mann–Whitney rank sum test with Bonferroni correction. (**A**,**B**) show exemplary light microscopic images at 400× magnification of TUNEL-positive cells in the control group (**A**) and ALM group (**B**) on day four after trauma induction. The cells marked in the circle represent TUNEL-positive cells. The scale in (**A**,**B**) corresponds to 25 µm.

**Table 1 biology-12-00870-t001:** Quantitative analysis of BrdU-, Ki67-, and TUNEL-positive cells of the right, healthy soleus muscle. After testing for normal distribution, an unpaired *t*-test was performed to examine significant differences in cell number between ALM groups and controls. Data are given as mean values ± SEM, * *p* < 0.05.

	BrdU-Positive Cells [n/mm²]		Ki67-Positive Cells [n/mm²]		TUNEL-Positive Cells [n/mm²]	
Day	ALM	Saline	ALM	Saline	ALM	Saline
1	11.4 ± 0.01 *	6.6 ± 0.93	5.7 ± 0.95	5.3 ± 0.19	0.8 ± 0.28	0.5 ± 0.09
4	11.5 ± 0.35 *	7.1 ± 0.63	10.6 ± 0.56	13.7 ± 1.62 *	0.3 ± 0.16	0.6 ± 0.23
7	8.6 ± 0.45 *	6.6 ± 0.38	11.6 ± 0.26	13.0 ± 0.65	0.6 ± 0.15	0.5 ± 0.14
14	7.2 ± 0.68 *	5.1 ± 0.39	18.5 ± 0.97 *	14.9 ± 1.41	0.4 ± 0.15	0.4 ± 0.01
42	3.8 ± 0.84 *	2.2 ± 0.27	17.2 ± 0.25	14.2 ± 0.65	0.6 ± 0.20	0.6 ± 0.12

## Data Availability

The data presented in this study are available on request from the corresponding author. The data are not publicly available due to privacy and ethics.

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
