# Peer review of "ALM Therapy Promotes Functional and Histologic Regeneration of Traumatized Peripheral Skeletal Muscle"

_biology, 2023, doi:10.3390/biology12060870_

Round 1

Reviewer 1 Report

This study examined a relationship between a single dose of the components of adenosine, lidocaine and Mg2+(A LM) iv and skeletal muscle regeneration after traumatic injury. However, seral problematic points were found in the method and basic skeletal muscle physiology.

Major:

1.         In my understanding, soleus muscle is innervated by the branches of tibial nerve on its surface of the mid portion as it should be observed in Fig. 1 and Fig. 2C. However, it could not be detectable in the state of Fig. 1B. So, I wonder, that sole muscle was denervated by the artificial Kocher clamp, although the authors likely also said the same matter in line 107-108. The branches are crushed and uprooted with made long gap. If so, this is a quite severe damage, and therefore, this model may be an excessive model for the short-term muscle regeneration experiment. It is questionable that the electrical stimulation induced muscle contractions occurred during 1-7 days after the damage.  

2.         Another point: the authors likely miss understanding the twitch contraction physiologically/biologically. The twitch contraction is composed just one EC (excitation-contraction) coupling induced by just one electrical stimulation. Thus, shortening phase shows the capacity/property of myosin head, and relaxation phase shows the capacity of sarcoplasmic reticulum. Therefore, the authors saying “fast twitches” induced by 75Hz to 0.1sec is not make sense for me. Actually, this is a short-time tetanus as same as the tetanus 75Hz /3sec.   

3.         A labeling time of BrdU for 48 hour is too long and unsuitable for this experiment. Labeling times 1-3 hours is much suitable. Because, day 1 group received 24h before muscle damage, as different from the others. This is specific and curious. Similarly, for the time course setting 1, 4, and 7, these time durations did not change a lot, because of the duration of 48h labeling. Additionally, I also incomprehensible to use combination BrdU and Ki67. In this study, it is enough only Ki67 staining.  

4.         For the handling of skeletal muscle samples, fresh and quick frozen by isopentane precooled by liquid nitrogen keeping in situ muscle length is very basic, and the best and correct way for the purpose of the morphological/histological analysis. Paraffin method after long time immersed 4% formalin fixations is a contraindication way, because it makes fiber shrinking, widely spread interstitial spaces with lower antigenicity. Unfortunately, the skeletal muscle physiologist (as I am/we are) cannot help rejecting the data from this method.

5.         The most problematic issue is why this study did not use myogenic marker staining such as Pax7, MyoD and myogenin. In addition, the authors also did not perform laminin staining for the basement membrane. In spite of that, the term “satellite cells” is frequently used. There is completely no proof of satellite cells in this study. In case of skeletal muscle traumatic damage, various progenitor/stem cells are induced for the proliferation, such as myogenic satellite cells, vascular endothelial cells, smooth muscle cells, pericytes, peripheral nerve relating Schwann cells, perineurial/endoneurial cells and multipotent interstitial stem cells together. There were all BrdU and/or Ki67 positive.          

Others:

Why the authors select the soleus muscle, which is composed mostly slow fibers, should be explained.

Figure 2 is somewhat grotesque, thus, probably it does not need. However, it is clear that the line between the distal tendon of the soleus muscle and transducer sagged and is too long, thus, I wonder that it was difficult to measure the correct tension in this condition. It should be improved.       

Author Response

Dear Reviewer,
Enclosed you will find the file with your comments and the corresponding answers. 

Best regards
Nina Hoeger

Author Response

Dear Reviewer,
enclosed you will find the file with your comments and the corresponding answers. 

Best regards
Nina Hoeger

Reviewer 3 Report

Comments

This team has published a couple of articles regarding experimental ALM therapy in survivable hemorrhage. This paper was to determine whether Adenosine, lidocaine and Mg2+ (ALM) has protective effects on traumatized peripheral skeletal muscle. Here, the authors have suggested that that traumatized skeletal muscle be positively affected by ALM therapy leading to increase muscle strength and cellular proliferation and to decrease apoptosis. They have been showing significant benefits on biomechanical force development and cellular behavior after ALM therapy. Some of the results are progressive, but too narrow to explain the mechanism or the mode of action implicated in ALM treatment. Little are new or novel as well as many gaps are behind their insistence. Current status of this article not is enough to publish.

Followed are major Concerns

1.     First of all, trauma mice treated with adenosine, lidocaine and Mg2+ (ALM) were recovered from twitch and tetanic in Fig 3. Individual effects of each adenosine, lidocaine and Mg2+ have been varified or mentioned? In addition, they need to decide whether there are any changes in the cross section of the regenerative skeletal muscle fibers from the traumatic area.  

2.     In order to insist muscle regeneration effects of the ALM, they need to show more evidence regarding muscle types in WT vs ALM mice including macrophage accumulation, type of macrophage, and size of muscle size etc.

3.     Please discuss or mention whether the ALM could effect on the age-related progressive loss of muscle mass and strength, sarcopenia.

Author Response

(The authors gave the same response as above.)

Reviewer 4 Report

Dear Authors,

Assess and amend the following minor defects:

Lines 37-38. The verb of the sentence is missed.  Amend it.

Line 92-. The age of animals must be mentioned. The age is a critical factor for muscle regeneration and the onset of sarcopenia. This might be properly addressed in the discussion. Similarly, regarding sex, although the study has been carried out with male rats, sex issues might be considered in muscle regeneration. This should be addressed in the discussion.  

Line 97-. How was the anaesthesia administered? Intraperitoneal? This must be stated.

Line 160 and other lines. Is it a 400x magnification? or the images were taken through a 40X objective? this should be clearly stated. Anyway, the scale bar in microscopic images would help to interpret the real magnification. Consider add the scale bar to images.

Line 165. More details should be written regarding the TUNEL method, how does it work? what detect?

Line 172. It should be “significance” instead “significances”.

Line 203. Improve redaction of sentence “... Indicated are mean values ± SEM, *p < 0.05.”

Line 245. Substitute the term “As much,”. It sounds colloquial.

Line 372. It must be written what EPO and G-CSF stand for, in addition to the acronym.

Author Response

(The authors gave the same response as above.)

Reviewer 5 Report

This manuscript describes the beneficial effect of ALM therapy on the regeneration from crush-injury of soleus of male Wistar rats. The effect such treatment is not entirely unexpected but the molecular and biological background is not entirely known so it is relevant to investigate. For this reason some improvement is suggested in order to make this work deeper.

Specific comments

163 – The role of BrdU is evident but Ki67 seven should be described and add reference, also this is not a satellite cell marker unless other markers (Pax 7) are used. Crushed muscle has inflammation and interstitial cell division on the first days and this give a strong background with 2nd Ab. What was the negative control for IH?

228 – The position of satellite cells (between the sarcolemma and basal lamina) is not enough for identification of myogenic cell divisions since only a fraction of satellite cells produces myoblasts, others can give rise to haematopoietic cells. (see PMID: 11776477 and PMID: 36152246 and others).

See the highest value of both BrdU and Ki67 positive cells on the first day which might imply inflammatory and other interstitial cell divisions as well.

247 – The largest difference in Ki67 + cells between injured and control muscles is on day 42, this shows what?

The data on Fig. 4 and 5 give a reason for concern since the histochemical sections of the soleus appear to be neither longitudinal nor transversal. This might have made difficult to compare the sections and raise the question would not it be better to count active nuclei per fibre instead of per mm2 ?

Author Response

(The authors gave the same response as above.)

Round 2

Reviewer 1 Report

Responses are fully unsatisfactory. 

1.         The reference below is filled with false messages in terms of the skeletal muscle physiology. Thus, the present study is a typical case of “the wrong message makes wrong study”. We need to stop the vicious circle. Actually, around 2006, a large number of papers showing the multiple differentiation of bone marrow-derived cells have been published, but there were mostly false, because of a lack of direct evidence associate with a lower quality of functional measurement. Today, it becomes that these are well-known facts, and confirmed that there is a significant decrease in these kinds of documents. The authors should know that a previously reported things not always right, and if wrong, they should be corrected hereafter.

Matziolis, Georg; Winkler, Tobias; Schaser, Klaus; Wiemann, Martin; Krocker, Doerte; Tuischer, Jens et al. (2006): Autologous Bone Marrow-Derived Cells Enhance Muscle Strength Following Skeletal Muscle Crush Injury in Rats. In: Tissue Engineering, 060303124145004. DOI: 10.1089/ten.2006.12.ft-41.

2.         Additionally, DOI information is also wrong.  As correctly, DOI: 10.1089/ten.2006.12.361

3.         I requested original responses of the authors. I do not want previous references. Show the raw data of twitches and tetanus. That is a respectable way. The duration of twitch is around 150 msec in rat soleus.

4.         About a satellite-like cells:   Actually, a term “satellite cell-like cells” is right. Then, quite honestly, only detection of proliferating cells without laminin (basal lamina) staining has not reached a level, which could be saying of satellite cell-like cells. That method is a 30-year-old way. The staining of Pax7, MyoD, CD31, p75 and skeletal/smooth muscle actin are the minimum requirement of the recent skeletal muscle regeneration study. It is obvious, and if not, it doesn't stand up the start line. Only paraffin section, too weak.     

5.         About the Fast-twitches:   Again, this is not a twitch. Current measurement is a tetanus induced by 75Hz/0.1msec duration, thus, this is a short time tetanus and what is physiologically significant? This is a text book level issue.

6.         In the point-by-point answers, basically, the author's current/original data should take precedence. For example, insert the newly attached photograph (innervation can be detectable certainly) in Fig. 1 and compared. It is clear that innervation has disappeared in Fig. 1B.

7.         It seems that the references overly reliant.

Reviewer 2 Report

Review #2

The manuscript was improved slightly but not enough for final acceptation.

I follow the numbering in the answer of the Authors.

Major point:

New issue: Thank you for increasing the size of histological images, but as a consequence of better visibility I had further objections. The Authors shows representative images of the soleus muscle sectioned longitudinally from proximal to distal. It is not the usual way to count proliferation markers on longitudinal images. This has to be justified in the Methods. Looking at the images in more detail, I have to say that not all of them look like longitudinal sections. Especially Figure 4B and 5B looks as a cross-sectional slice. Since it is missing from the Methods I suppose that black dots were counted in all images. The way of counting should be given in the Methods. It would be helpful, if the Authors show images at the end of the experiment (day 42, new panels in Figure 4-6) to show how the number of positive cells changes. Please see Stratos et al., AJP 182, 2013.

Comment 2:

Methods: The exact composition of ALM solution given in the answer of the Authors should be inserted in the Methods.

Comment 3:

One of the listed references in the response should be inserted in row 165 for BrdU treatment. The difference in “Day 1” group treatment with BrdU should be mentioned in the Methods.

Comment 4:

The description of the in vivo muscle measurement should be extended with the text in the answer of the Authors.

It is pity that the Authors did not document the change of injured muscle by photos. It is a really shortcoming of the manuscript. The Authors should prepare a supplementary figure showing the localization of the penumbra.

The width of tissue sections used in histochemistry is still missing and should be given.

Comment 5:

Results: Please prepare a figure from the original twitch and tetanus traces you showed in the answer (Response 5).

I still not understand how the values of twitch (0.55 N) and tetanic (0.91 N) force from healthy soleus were calculated and used for the normalization.

I insist on deleting the “fast” from “fast twitch” term. I accept that there are papers in the literature (cited by the Authors in the answer), which uses this term, but 99% of the papers use the term “fast twitch” as a property of a muscle. Please follow the nomenclature used by the majority of the scientific community.

Comment 6:

Please prepare a supplementary table containing the number of BrdU-, Ki67-, and TUNEL-positive cells in the control (not injured) soleus (Response 6). It is not clear why the Authors used different labeling to show significant differences in Figure 6. Please unify the labelling.

Minor point:

Comment 17:

Please insert the label of scale bar of images in Figure 4-6 into the figure legend, because they are too small. Please label with letter “C” the column graphs.

Reviewer 5 Report

Answers to comment 1 are acceptable but they have to be introduced into the modified version of the manuscript, including the additional references in order to improve the manuscript.

Answers to comment 2 have to be also included in the modified version with key references cited in the comment. The "satelite-like cells" is not an appropriate term in my opinion because once they have been identified as satellite cells by position. If these cells are labelled as proliferative ones but it is not sure if give rise to myoblast or not is an other matter.

Answers to comment 3 are not acceptable. I firmly believe that it would be better to count satellite cells per fiber cross section. It should be possible to find to make similar cross sections from treated and control muscleseEven if the muscles are “traumatised” (being in contracted state?)

Round 3

Reviewer 1 Report

Obviously, the quality of this manuscript has not improved at all, because the authors still do not understand the basic biology and physiology of muscle contraction, and the methodology of immunohistochemistry.  

1.         Authors still written that “For twitch stimulation with 9 mA / 75 Hz bipolar was applied five times for 0.1 s each time at an interval of 5 s. For tetany stimulation was applied at 9 mA / 75 Hz five times for 3 s each time at an interval of 5 s” (lines 152-154), and looking their response, it is obvious that the authors still fail to understand the "twitch" contraction. This is a very basic biology level issue that the twitch is a contraction phenomenon induced single stimulation induced single E-C coupling. In consideration of a mechanical fatigue of E-C coupling, it should be taken 1 sec or more interval between stimulation. Think the duration of twitch in soleus. Thus, 75H/0.1 sec stimulation is short-time tetanus, and they still never respond the reason for this analysis. By their response, unfortunately, it seems that the authors’ group has failed to understand this issue since more than 20 years ago. Of course, this is unacceptable and should never appear in the current scientific paper. 

2.         Figure 3 legend “(modified according to Stratos)” what does it mean?

3.         Additionally, what is the point of Figure 3? The readers want to see the data comparison among the traumatized ALM treated and non-treated group, and healthy control. Thus, this is missing the point.

4.         Similarly, Figure 4 is also unnecessary, and caused the violation (see below).

5.         What kind of staining has been done in Figures 4, 6 and 7?

6.         I suppose that there are hematoxylin & eosin (HE, if there were toluidine blue, not helpful, see below), because of the paraffin section. If so, how obtained the reaction product of BrdU and Ki67? There were probably DAB after using first (in case of BrdU) and secondary antibody HRP conjugated. Under above condition, reaction product of DAB wholly masked the unclear staining of hematoxylin. This is an old-fashion method (about 3 decades gap), and in this way, HE staining should have been performed very weakly. However, the authors have done strong HE staining, thus, in facts, it is hard (impossible) to identify the differences between nuclei in the circle and the others. Therefore, unacceptable presently (currently widely using fluorescence immunohistochemistry with Ki67 or PCNA).

7.         In the above regard, it is natural to correspond with the results of BrdU and Ki67, and explanation of the results of both staining is not necessary. In addition, from the view point of “reduction of animal pain”, it was not necessary the BrdU analysis. (I have raised the usage of BrdU at first time, thus should be awarded) Therefore, this can be an unacceptable reason like an unnecessary pain given to the animal. If the authors' group has been generally used for this method, it is matter.   

8.         The authors used the same photograph in Figure 4A and 6A, but no explanation provided. Basically, this is one of critical violation, thus unacceptable. It should be used the other one.  

9.         “The proliferating satellite cell-like cells were located between the basal lamina and the muscle fiber” How did the authors detect basal lamina? Did you perform laminin staining? No information presented, thus, no proof and unacceptable.

10.     Figure 5 still using “twitch force” in the ordinate. Anyway, the results after 14 days are no differences. Thus, it is possible that there are no differences in therapeutic meaning.

11.     In regard to the above comment 6; it is quite difficult to identify positive/negative reactions of BrdU in Figure 6A and B. Thus, the readers just impossible to image the next results of graph C.  

12.     I have suggested in the previous comments that it is hard to analyze the muscle regeneration after trauma using only the detection of proliferating cells. For example, functional significance was observed during 7 days after operation in the ALM group, however, the relative number of proliferating cells were also higher in the ALM group. How explains these results?

13.     Basically, a higher number of proliferating cells in the muscle shows a higher number of activation/proliferation of various cells (myogenic, peripheral nerve, vascular cells and interstitial stem cells with the myofibroblasts and satellite cells). In addition, induced trauma relating immunological cells such as neutrophil and macrophages also aggregated and must be appeared as mononucleated cells in photographs. This is also the basis of current skeletal muscle physiology. However, this study never explained and proved a piece of these mechanisms.

14.     After the severe muscle trauma, first 7 days are inflammation term and real muscle regeneration began after 10-14 days following the increase in amino acid uptake with acceleration of myosin/actin formation. Did you detect first 7days? No proof.

15.     Did you detect the increase of the formation of muscle contractile component? There is no proof.

16.     Did you used satellite cell marker Pax7? No proof again.

17.     Similarly, did you detect CD31, p75, vascular smooth muscle, NG2, laminin, MyoD and N200? Completely no proof.

18.     Current paper of muscle regeneration routinely performing several of them minimally.

19.     Actually, did you do a comparison of the administered ALM in the muscle during 7 days and at late 14 and 42 days of ALM group? Did you try to detect increased PLA in op group muscle or blood samples? All answer should be no.

20.     On the other hand, the explanation of the condition of BrdU injection, particularly 1 day group modified from first draft and response, without reports in the response. Violation?  

21.     Consequently, recovery of tetanic functions did not change between both operated and control groups, and at 14 and 42 days the rate reached over 80% recovery. Therapeutic potential is not so reliable enough. 

22.     This paper is likely a back to the time old paper to the late 20th century, but now this kind of old fashion paper could not hold the starting line.  

23.     One thing, this is old fashion paper but strict description role of the old paper method did not do, especially immunohistochemistry of BrdU. This including DNA denaturation method and wash well, quality of secondary anti-body, and what kind of color selected with containing substance, probably laminin.

24.   Above all, I cannot find out the acceptable point of this paper. In my impression and consideration, the authors should study muscle physiology paper and the muscle regeneration paper over the groups. Discussion of the same groups could not get any other new idea. 

Reviewer 2 Report

Review #3

The manuscript was improved sufficiently.

I follow the numbering in the answer of the Authors.

Minor point:

Comment 5:

The Authors did not understand my problem with the values of twitch (0.55 N) and tetanic (0.91 N) force from healthy soleus. I suppose that these values are averages from several (n should be given) healthy soleus muscles. If I am right, they have errors. It is OK, that for normalization the errors do not used, but still they have to be given in the text. If my assumption is not correct, please explain why you used these values for normalization.

Comment 6:

Thank you for the table containing the number of BrdU-, Ki67-, and TUNEL-positive cells in the control (not injured) soleus (Table 1). Please indicate that the unit is n/mm2. Please add a sentence about the statistical analysis (I hope you did). You should compare the ALM and C columns in Table 1.
